# Graphene Oxide-Chitosan Aerogels: Synthesis, Characterization, and Use as Adsorbent Material for Water Contaminants

**DOI:** 10.3390/gels7040149

**Published:** 2021-09-24

**Authors:** Filippo Pinelli, Tommaso Nespoli, Filippo Rossi

**Affiliations:** Department of Chemistry, Materials and Chemical Engineering “Giulio Natta”, Politecnico di Milano, 20131 Milan, Italy; filippo.pinelli@polimi.it (F.P.); tommaso.nespoli@mail.polimi.it (T.N.)

**Keywords:** aerogels, chitosan, graphene oxide, water treatment, dyes removal, electrostatic interactions

## Abstract

Porous aerogels, formed by subjecting precursor hydrogels using a freeze-drying process, are certainly one of the most studied and synthetized soft materials, thanks to their important features such as elasticity, swelling behavior, softness, and micro and nanosized pores, which guarantee their applicability in various fields. Typically, these systems are synthetized working with natural or synthetic polymers, but in the last years great interest has been given to proper formulated aerogels able to combine polymeric structures with other moieties such as graphene or graphene oxide. This working strategy can be pivotal in many cases to tune important properties of the final system such as toughness, porosity, elasticity, electrical conductivity, or responsive behavior. In this work we propose the synthesis of chitosan graphene oxide aerogels obtained through self-assembly of graphene oxide sheets and chitosan chains. These three-dimensional systems were chemically characterized with IR and XRD technique and their inner structure was investigated through the scanning electron microscopy (SEM). Moreover, we mechanically characterized the material through dynamic mechanical analysis, showing the stability of these systems. Finally, the adsorption ability of these soft materials has been demonstrated using model molecules to simulate water contaminants showing the efficacy of those graphene-based systems even for the removal of anionic dyes. Complete removal of contaminants was obtained at low concentration of dyes in solution (100 mg/L), while with a higher amount of pollutant in the solution (350 mg/L) high sorption capacity (q > 200 mg/g) was observed.

## 1. Introduction

Hydrogels are nowadays widely recognized as one of the most versatile and flexible biomaterials and their uses range from biomedical applications, such as drug delivery or tissue engineering, to sensors and environmental applications such as water decontamination [1,2,3,4]. This important versatility, verified by the great attention that hydrogels are attracting, is guaranteed by the unique features of this kind of three-dimensional polymeric network, such as hydrophilicity, swelling behavior, softness, and presence of micro and nano pores [5]. In the last years, various strategies have been developed to improve some of the characteristic of these networks, such as the development of double network hydrogel or the incorporation of different moieties in the system [6,7]. In this context, graphene composite hydrogels certainly represent a very interesting strategy, able to combine the exceptional features of graphene with the characteristics of this kind of gel [8]. Graphene and its derivatives are nowadays attracting the interest of researchers from different fields thanks to their multiple properties such as conductivity, but also thermal and chemical stability and mechanical properties [9,10]. Graphene oxide (GO), a precursor of chemically converted graphene, has been successfully employed in the development of hydrogels framework exploiting the presence of epoxides and carboxyl groups that favor its aqueous dispersion and make this material suitable for combination with polymeric chains and hydrogel framework synthesis [11,12]. The combination between GO and proper polymeric chains guarantees the possibility to encapsulate and capture various molecules exploiting different kind of interactions such as van der Walls interactions but also hydrophobic or electrostatic. In fact, oxygen groups can bind metal ions and positively charged organic compounds through coordination or electrostatic interaction [13,14]. On the other hand, it is very important to highlight that when subjected to freeze-drying process, hydrogels lose their water content, and their structures are commonly regarded as aerogels [15]. These materials obviously maintain their chemical composition and still present high porosity, large specific surface area, and affinity with various molecules and they are therefore suitable for various applications such as support of catalysts, absorbents of dyes, and sensors [16,17]. In this work we propose the synthesis of graphene oxide chitosan (GO-CS) aerogels obtained through self-assembly of graphene oxide sheets and chitosan chains into precursor hydrogels and subsequent lyophilization. This process allows us to combine the excellent properties of chitosan, such as biocompatibility and biodegradability, with graphene oxide [18]. Moreover, the addition of chitosan can balance the negative charge of GO and guarantee the possibility, for the final framework, of capturing anionic compounds via electrostatic interactions improving the adsorbent properties of this kind of gels [13]. These kinds of systems have been already reported in the literature and the tendency of GO to self-assemble with CS chains in acid solution has been studied as a possible strategy to enhance the mechanical stability of chitosan aerogels [19,20]. Unfortunately, CS/GO aerogels are relatively unstable and have the tendency to dissolve in weak acid conditions [21]. Because of this, physical and chemical crosslinking processes have been adopted to produce robust CS-gels [22]. The crosslinking reaction involves the functional groups of the CS chains, reducing the number of active sites in the adsorption process and preserving the mechanical and chemical stability of final gels [23]. In the last years new strategies have been adopted for the synthesis of these systems. The use of ultrasound irradiation at different stages of preparation and application of the material have been successfully adopted to enhance the adsorption efficiency of CS/GO nanocomposites [24], while graphene oxide reinforced chitosan systems with chemical interfacial bonding have been produced through the amide reaction that significantly improved the interfacial interactions and the mechanical properties of the composites [25]. Moreover, the synthesis of supramolecular hydrogels of chitosan and GO have been reported in literature, with the self-assembly of chitosan chains with graphene oxide nanosheets that works as the two-dimensional crosslinker of the composite material exploiting noncovalent interactions [26].

In this present work we evaluated various formulations and ratios between chitosan and graphene oxide, employing the ammonium persulfate (APS) as crosslinking agents for the polymeric framework. We consider the ease of synthesis, in term of gel formation, and the shape of the inner framework investigated with SEM [27], and, finally, we focused on a single aerogel formulation, deepening various aspects and features of the material through chemical and mechanical characterization. We tested the sorption efficiency of this formulated aerogels, working with two organic dyes Indigo Carmine (IC) and Cibracron Briliant Yellow (CBY), which present similar structures but different molecular weights and number of sulphonate groups, which are important parameters for organic dyes and their environmental impact [28]. The analysis, realized at room temperature and neutral pH, confirmed the efficacy of the synthetized aerogels as absorbent materials.

## 2. Results and Discussion

### 2.1. Aerogels Formation and Characterization

The aerogel network was synthetized through the cross-linking of graphene oxide sheets through chitosan chains. In fact, graphene oxide can be easily dispersed in water thanks to the electrostatic repulsion between their sheets, and the addition of any reagents able to reduce the repulsions or increase the bonding forces of the system leads to an instability of the dispersion and to a physical change in the system [29,30]. In our case, working with chitosan, a positively charged polymer, we introduced aminic groups able to strongly attract the graphene oxide balancing its negative charges. Moreover, the dissolution of chitosan in the acidic medium guarantees the formation of cationic polymers, through the protonation of the amino groups of the material. Therefore, the electrostatic interactions between the positive charges of chitosan and the negative ones of the functional groups of graphene oxide are responsible for hydrogen bonds genesis and the precursor hydrogel formation [31]. The schematization of the synthesis method and the applications of the aerogels in water purification is reported in Figure 1.

The inner structure of the aerogels, obtained from the lyophilized hydrogels, was investigated through scanning electron microscopy as reported in the following Figure 2.

The typical SEM image of lyophilized hydrogels is reported, with a clear tridimensional porous structure with pore sizes in the order of micrometers. In particular, using a published protocol we obtained a value of porosity around 75% [32]. No polymer particles are visible in the framework and a uniform distribution of both the constituents can be observed confirming the efficacy of the mixing procedure employed. Often, in fact, for this kind of formulation, issues related to excessive viscosity of the system can lead to a non-homogeneous network with inner defects in the structure [31].

The synthetized material was chemically characterized through ATR-FTIR analysis and with the investigation of its XRD pattern, reported in Appendix A. In the following Figure 3 we reported the spectrum of the synthetized material as transmittance [%] versus wavenumber [cm^−1^]. Characteristics of the spectra of chitosan and graphene oxide can be detected in the spectra of GO-CS aerogels with variations in intensity and shift towards higher wavenumber due to the interaction between the polymer and the graphene oxide. These features can be observed especially for peaks at ~1600 cm^−1^ (**2**), attributed to C=O stretching and around ~1066 cm^−1^ (**3**) corresponding to C-O stretching. Moreover, a band at 1589 cm^−1^ corresponds to the N-H bending of the primary amine, but it was probably overlapped by other bands in this spectrum [33]. Finally, an additional wide band for the composite material was observed between 3000 and 3400 cm^−1^ (**1**) due to the O-H functional groups of graphene oxide [34]. Thermogravimetric analysis (TGA) was used for the stability of aerogels (Appendix A). Weight loss was observed at a temperature range from 41 °C to 92 °C attributed to the removal of water molecules present inside the aerogels. Weight loss of about 50% was observed around 250–485 °C which may be attributed to the pyrolysis of the lesser grafting of CS contained in CS-GO aerogels.

### 2.2. Mechanical Characterization of Lyophilized Gels

The synthetized GO-CS composite aerogels were mechanically analyzed using both compressions tests and dynamic mechanical analysis. As previously reported in Section 4, compressions analysis was performed in the region between 0% and 70% of maximum extensional strain and in Figure 4 the corresponding compressive stress-strain curve is presented. The analyzed aerogel presents the typical behavior of porous materials: in the initial region of the curve a confined region with the typical trend of the elastic deformation behavior of the material can be observed, followed by the subsequent plastic deformation region [35]. After this, it can be observed how the strain of the materials increases without significant variations in the values of compressive stress. This is due to the progressive collapsing of the internal porous structure of the material during the compression of the specimen in the test that led to a densification of the system. When the sample is completely densified, the material exhibits the typical behavior of non-porous material, and the stress values show rapid increase until maximum extensional strain (70%) [36]. The maximum values of stress, obviously reached in correspondence of the maximum strain, are considerably good and demonstrate the mechanical stability of the synthetized system. From the comparison with the mechanical properties typical of neat chitosan hydrogel it is well evident how the presence of GO inside the framework can guarantee an improvement in the characteristic of the final device [37]. This feature can be observed considering the higher value of stresses that can be applied on the sample in correspondence with the same strain value.

### 2.3. Adsorption Tests

The adsorption kinetics tests were conducted using two solutions of different known starting concentrations for IC and CBY: C_0−A_ = 100 mg/L and C_0−B_ = 350 mg/L. The time-dependent adsorption curves of those organic dyes are reported in the following figures as sorption capacity q versus time, Figure 5, and as percentage of dye removed versus time, Figure 6 [38]. It is well evident how, for both dyes and for both concentrations, it is possible to observe fast adsorption kinetics. The sorption capacity values of the materials confirmed the efficacy of those systems [39]. Working with C_0−A_ and with the conditions described in the Section 4, values around 70 mg/g of adsorbed dye per mass of adsorbent material were obtained very quickly for IC and CBY and in both cases they were sufficient for an almost complete removal of dyes. On the other hand, working with C_0−B_, the sorption capacity of GO-CS aerogels for IC constantly increases in time, reaching a plateau value around 220 mg/g of adsorbed dye per mass of adsorbent material. Even with CBY the sorption capacity increased in time, reaching lower values around 180 mg/g. This difference can be attributed to the smaller dimensions of the molecule of IC (MW_IC_ = 466.35 g/mol, MW_CBY_ = 831.02 g/mol) that favor its penetration inside the structure of GO-CS aerogels [40].

Considering the percentage of removed dyes, 100% of IC was adsorbed in the first 20 min, while more than the 90% of CBY was adsorbed in the same time interval, confirming the efficiency of this material. On the other hand, when working with a higher concentration (C_0−B_), it takes around 75 min to reach 90% of dye removal for IC, while in the same time interval, only 50% of CBY was removed. In both cases, it is well evident how the adsorption gradually increases with the contact time. Comparing those results, it emerges how IC presents a better kinetics than CBY, again probably due to the smaller dimensions of the molecule.

The adsorption of the dyes can be explained considering the chemical structure of the materials constituting the aerogels [13]. Graphene oxide, thanks to its large surface area and its negative charged sites, has strong affinity with cationic dyes but it cannot be employed with anionic ones. The combination with chitosan and its positively charged chains determines in the final composite materials the ability to remove even anionic dyes, like those employed in this work, through electrostatic interactions. This is a very important feature of our material since graphene oxide has been commonly employed in the removal of cationic dyes, exploiting its negative charged sites. Its combination with chitosan enlarges the possibilities for use of graphene oxide, which is certainly an ideal adsorbent material thanks to its features.

## 3. Conclusions

In this work we successfully synthetized and characterized graphene oxide-chitosan aerogels and we demonstrated their efficacy for the removal of organic dyes from aqueous solution. Working with solutions of 100 mg/L of pollutant, CS/GO gels were able to completely remove the contaminants with sorption capacity (q) of 70 mg/g. In case of higher concentration of the pollutant in the solutions (350 mg/L) an almost complete removal of the dye was observed for IC (>90%), while only a partial removal for the CBY has been obtained (50%), as has been discussed in detail in the previous section. The electrostatic interactions between the positive charges of chitosan and the negative ones of the functional groups of graphene oxide, responsible for hydrogen bonds genesis and the precursor hydrogel formation, were demonstrated to be an efficient strategy to produce this kind of system. SEM analysis confirmed the formation of a soft and highly porous structure that is pivotal for adsorbent material. This feature has been demonstrated by mechanical characterization, where the aerogels showed the classical behavior of porous materials. Finally, the adsorption tests with organic dyes indigo carmine, and cibacron brilliant yellow, showed high values of sorption capacity for these materials and confirmed their efficacy in the water purification and the applicability of graphene oxide-based system even for the removal of anionic dyes.

## 4. Materials and Method

### 4.1. Materials

Graphene oxide dispersion (10 mg/mL) was purchased from GOgraphene, trading name of William Blyte Limited (Harlow, Essex, England), while chitosan (low molecular weight) was purchased from Sigma-Aldrich (Sigma-Aldrich Chemie GmbH, Deisenhofem, Germany). All other used chemicals were bought from Sigma-Aldrich (Sigma-Aldrich Chemie GmbH, Deisenhofem, Germany). The materials were used as received; solvents were of analytical grade.

### 4.2. Synthesis of Graphene Oxide-Chitosan Composite Aerogels

The composite hydrogels made of chitosan and graphene oxide were synthetized using an acidic medium. Chitosan was dissolved in 2 mL of 2.5% *v*/*v* aqueous acetic acid and after 2 h the graphene oxide aqueous dispersion was added in the same vial. Then ammonium persulfate, as oxidant agent to promote the polymeric crosslinking, was dissolved in 0.5 mL of water and added to the system to promote the formation of the network. The hydrogels were frozen at −40 °C and then lyophilized to obtain final aerogels. Various ratios between chitosan and graphene oxide were employed; finally, due to the ease of the formation of the framework and some preliminary analysis, we decided to work with the ratio 1:1.7 between graphene and chitosan.

### 4.3. Materials Characterization

All the synthetized aerogels were characterized through scanning electron microscope (SEM) to investigate the shape and the porosity of the inner framework of the materials. SEM analyses were realized using a Zeiss Evo50 with EDS Bruker Quantax 200, while the aerogels were chemically characterized with attenuated total reflection (ATR) analysis, using a Varian 640–IR spectrometer from Agilent Technologies. Spectra were collected at room temperature under dry nitrogen atmosphere in the 400–4000 cm^−1^ wavenumber range, with an average of 64 repetitive scans to guarantee a good signal-to-noise ratio and high reproducibility and with a resolution of 4 cm^−1^. Moreover, XRD pattern of the material was obtained with Philips x-ray diffractometer model PW1830 (Kα1Cu = 1.54058 A°). TGA analysis was done by using SDT Q600 (TA Instruments) at a scan rate 10 °C/min under nitrogen flow (100 mL/min).

### 4.4. Mechanical Analysis

The synthetized graphene oxide-chitosan aerogels were mechanically characterized using an Anton Paas MCR 702 TwinDrive Rheometer equipped with 25 mm plane plate, using a compression setup. The analyses were realized on the lyophilized gels (aerogels) with cylindrical shape and dimensions of 9 mm of diameter and 5 mm of thickness. Two different tests were realized: first, static compression was performed, to obtain the stress-strain curves for the samples. The data were collected applying a 100 Pa preload, followed by a compression a 1 mm/min until the maximum extensional strain of −70% was reached. The viscoelastic properties of the materials were evaluated through the dynamic mechanical analysis. The tests were carried out under controlled strain, varying the amplitude in the range 0.1–10% with a logarithmic ramp. A preloading force of 0.03 N ad a frequency of 1 Hz were applied. All the tests were performed at 20 °C.

### 4.5. Adsorption Tests

To verify the efficacy of the synthetized aerogels as adsorbent materials we chose two different organic dyes characterized by different molecular weight and different number of sulfonate groups in their chemical structure: Indigo Carmine (IC, MW = 466.35 g/mol, λ_max_ = 610 nm) and Cibacron Brilliant Yellow (CBY, MW = 831.02/mol, λ_max_ = 402 nm). We prepared two different solutions of the two dyes of known concentrations (100 mg/L and 350 mg/L), and we immersed the lyophilized hydrogels in these systems, keeping constant the ratio between the mass of adsorbent material and the solution equal to 1.33 mg/mL. The samples were shaken, and we analyzed, at specific time intervals, aliquots of the solution, each time taking 1 mL of it which was then put back into the system after the analysis to avoid altering the system. From the absorbance values measured, through the calibration lines, the trend of the pollutant concentration over time was obtained. The sorption capacity *q*, corresponding to the mass of adsorbed dye per unit mass of adsorbent material, and % dye removed were evaluated each time with the following formula:(1)$$q=\frac{m_{0}-m_{t}}{m_{XG}}$$
(2)$$\%\;dye\;removed=\frac{m_{0}-m_{t}}{m_{0}}$$
where *m*_0_ and *m_t_* are the masses in mg of the dye pollutant in the volume of solution at the beginning and at time *t*, respectively, and *m_AG_* is the mass in g of the aerogel.

### 4.6. Spectroscopy Analysis

Quantitative spectrophotometric analysis allows the quantification of the concentration of a certain substance by measuring the absorption of UV-vis radiation by the molecules. Data were recorded on a V-600 Series UV-vis spectrophotometer from JASCO (Cremella (LC), Italy). Calibration lines were obtained recording absorbance values of known concentration solutions, and those profiles are reported in the Appendix A. The solution to be analyzed, placed inside a 1 cm × 1 cm base cuvette, absorbs an incident radiation with a selected wavelength equal to the characteristic λ_max_, obtained from the UV-vis spectra of the two dyes, while a detector measures the intensity of the radiation exiting the sample [28]. The absorbance of the sample is calculated by the software using the following formula:(3)$$A=\log_{10}\frac{I_{0}}{I}$$
where *A* is the absorbance and *I*_0_ and *I* are the intensities of the incident radiation and of the exiting radiation from the sample. The absorbance of a sample and its concentration are linearly correlated through the Lambert-Beer law (Equation (4)), which is valid only at high dilutions.
(4)A=ε C l
where *C* is the molar concentration of the sample, *l* is the optical path in cm, negligible for a 1 cm cuvette, and ε is the molar extinction coefficient that is characteristic of each substance and represents the absorbance of the sample at unitary concentration and unitary optical path.

## Figures and Tables

**Figure 1 gels-07-00149-f001:**
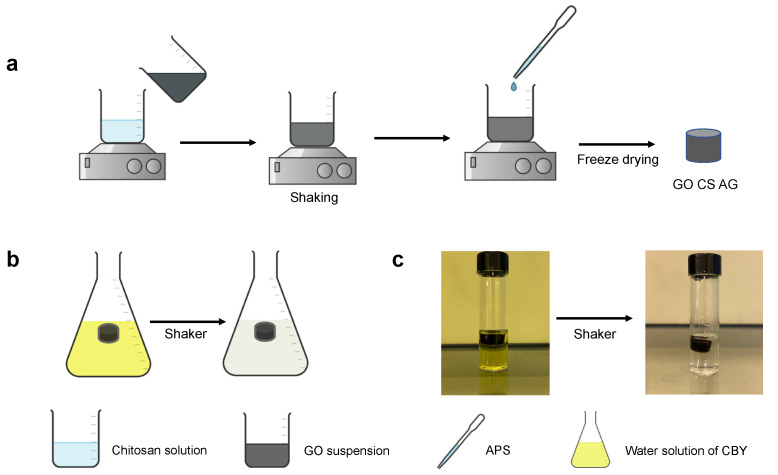
(**a**) Schematization of the synthesis method for GO-CS aerogels: GO suspension is added, under vigorous stirring, to chitosan aqueous solution. The addition of APS as oxidant agent promotes the crosslinking of the polymeric framework and the final gel is recovered through freeze drying. (**b**) Schematization of the removal of organic dye from water solution working with the synthetized aerogels. (**c**) Real application of the GO-CS AG in the removal of CBY from aqueous solution (C_0_ = 100 mg/L, shaking for 20 min).

**Figure 2 gels-07-00149-f002:**
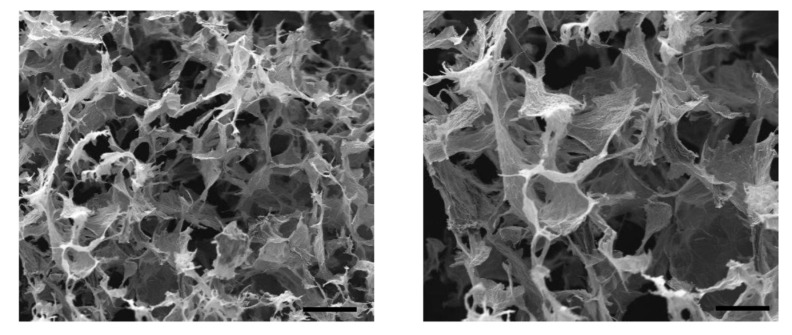
SEM images of the chitosan-graphene oxide composite aerogels. Scale bars: 100 μm for the figure on the **left**, 10 μm for the figure on the **right**.

**Figure 3 gels-07-00149-f003:**
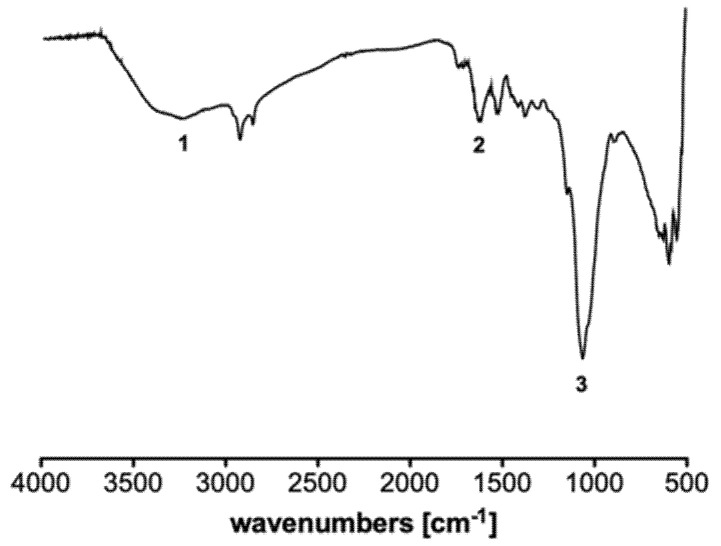
ATR-FTIR spectrum of the GO-CS composite aerogels in the range 4000–400 cm^−1^. (1) O-H stretching; (2) C=O stretching, (3) C-O stretching.

**Figure 4 gels-07-00149-f004:**
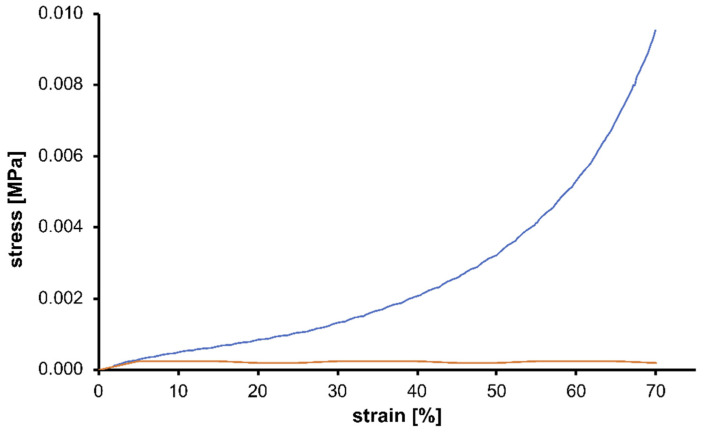
Compressive stress-strain curve for the lyophilized GO-CS aerogel (blue line) compared with the same sample obtained without GO (red line).

**Figure 5 gels-07-00149-f005:**
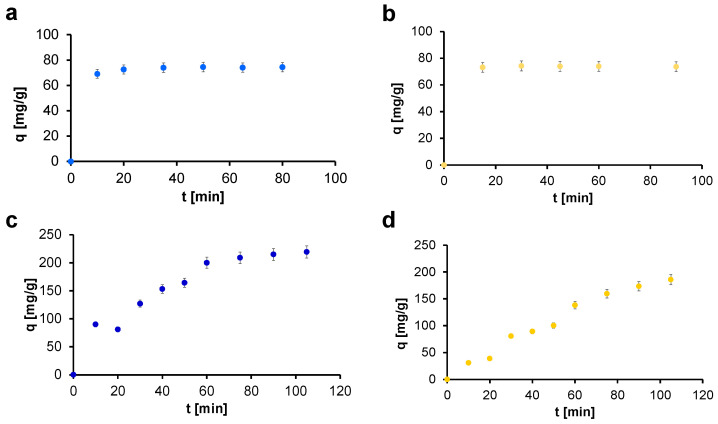
Sorption capacity of GO-CS AG to IC (blue dots) and CBY (yellow dots) in different concentrations: (**a**) IC 100 mg/L, (**b**) CBY 100 mg/L, (**c**) IC 350 mg/L, and (**d**) CBY 350 mg/L.

**Figure 6 gels-07-00149-f006:**
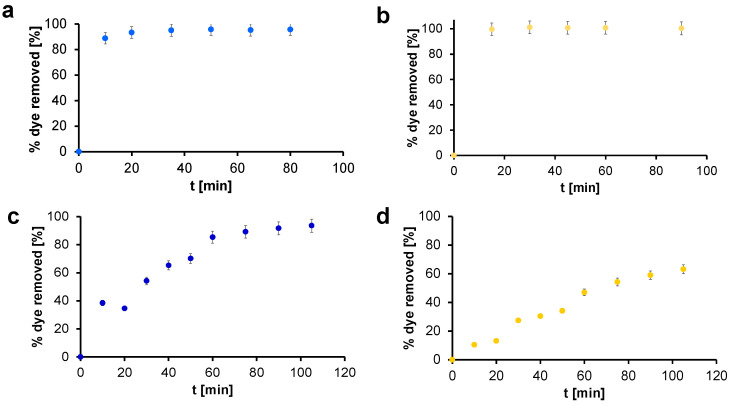
Percentage of removed dyes for GO-CS AG with IC (blue dots) and CBY (yellow dots) in different concentrations: (**a**) IC 100 mg/L, (**b**) CBY 100 mg/L, (**c**) IC 350 mg/L, and (**d**) CBY 350 mg/L.

## Data Availability

The data generated or analysed during this study are available from the corresponding author on reasonable request.

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
