# Peer review of "Graphene Oxide-Chitosan Aerogels: Synthesis, Characterization, and Use as Adsorbent Material for Water Contaminants"

_gels, 2021, doi:10.3390/gels7040149_

Round 1

Reviewer 1 Report

The paper presents an interesting approach on the synthesis of chitosan/graphene oxide xerogels and their application as absorbers of water contaminants.

This work, however, has substantial and technical/presentation deficiencies as follows by reference to line-numbering and of quoted importance:

  1. Line 10 – 15. Quotation on porosity and other merits are not backed up by experimental evidence.
  2. Line 18. No XRD data is included in manuscript.
  3. Line 19. Dynamic mechanical analysis evidence is not adequate (see also comments in text)
  4. Line 59 onwards. Chitosan is capable to capture anionic compounds as opposed to cation affinity of GO. This creates substantial questions on the validity and use of the concept presented here, which clearly not unjustified. No comparative investigation between pure chitosan and chitosan-GO synthesized here was made. If the authors had claimed specificity of absorption, they could have produced respective experimental evidence.
  5. Line 64. ‘SEM’ already defined to be used/ very informal writing: ‘end of the day’... 
  6. Line 68. No abbreviation of dyes is provided here, but is used  in text.
  7. Line 80. Undefined moieties.
  8. Line 90 and fig 1. Not properly described procedures.
  9. Line 107 onwards. The ATR or other analysis do not provide specific evidence of the presence of GO and possible connectivity (if any) to chitosan. Fig 3 not well drawn -with axis missing.
  10. Line 122 onwards and Fig 4. Description and justification are not evidenced in the findings of Fig 4. This makes relevant statements highy questionable.
  11. Line 139 onwards. Undefined abbreviations.   
  12. Line 140 onwards. Undefined experimental procedures, with no clear evidence given in Supplementary Information. No spectrophotometer data is provided.
  13. Line 155 /166 – Fig. 5-Fig. 6. More descriptive legends are needed on graphs to aid the reader.
  14. Line 170 and 185. No porosimetry data is provided.
  15. Line 194. Missing data of materials used.
  16. Line 211 onwards. Data provided is incomplete.
  17. Line 219 section. Data provided is incomplete and not evidenced in Fig. 4
  18. Line 231 section. The methodology followed is not properly described. No calibration data is mentioned. No spectra is provided,…
  19. Line 256. Spectra is mentioned in many sections but, neither correlations stated, nor examples are provided thereof .
  20. General comment: Language and styling need serious revision.  Informal phrases and repetitive sentences like 'thanks...'

Author Response

Reviewer 1

The paper presents an interesting approach on the synthesis of chitosan/graphene oxide xerogels and their application as absorbers of water contaminants.

This work, however, has substantial and technical/presentation deficiencies as follows by reference to line-numbering and of quoted importance:

Line 10 – 15. Quotation on porosity and other merits are not backed up by experimental evidence.

Porosity can be proven considering SEM analysis as presented in literature studies (Elia P. et al. Micropor. Mesopor. Mat. 2016, 225, 465-471 and Anguiano M. et al. Plos One 2017, 12, e0171417). Indeed from the images two different zones are visible: white fibers that constitute the network and black holes that represent the void spaces. We added the details in the manuscript.

Line 18. No XRD data is included in manuscript.

Sorry for the inattention. XRD spectrum is now added in Supporting Information.

Line 19. Dynamic mechanical analysis evidence is not adequate (see also comments in text)

The amelioration using GO together chitosan is visible from mechanical analysis. Indeed from the comparison with the mechanical properties obtained from the same system without graphene oxide it is well evident how the presence of GO inside the framework can guarantee an improvement in the characteristic of the final device (Gong Y. et al. Polymers 2019, 11, 777). This feature can be observed considering the higher value of stresses that can be applied on the sample in correspondence of the same strain value (now added in Figure 4).

Line 59 onwards. Chitosan is capable to capture anionic compounds as opposed to cation affinity of GO. This creates substantial questions on the validity and use of the concept presented here, which clearly not unjustified. No comparative investigation between pure chitosan and chitosan-GO synthesized here was made. If the authors had claimed specificity of absorption, they could have produced respective experimental evidence.

In this work we investigated the role of adding GO in chitosan aerogels and in particular their use in dye removal from water. The addition of GO is able to increase the mechanical properties obtaining a more stable device, fundamental for the future industrial use.

Line 64. ‘SEM’ already defined to be used/ very informal writing: ‘end of the day’... 

Thank you, we changed and used here the term ‘SEM’. Moreover, we revised the manuscript to avoid informal writing.

Line 68. No abbreviation of dyes is provided here, but is used in text.

Thank you, we provided abbreviations here.

Line 80. Undefined moieties.

In that sentence the term “moieties” was used to talk about the aminic group of chitosan. We modified the sentence to make it clearer.

Line 90 and fig 1. Not properly described procedures.

Thank you, we added a couple of sentences to clarify the procedure in the figure description.

Line 107 onwards. The ATR or other analysis do not provide specific evidence of the presence of GO and possible connectivity (if any) to chitosan. Fig 3 not well drawn -with axis missing.

Based on various works reported in literature we identified in our spectra various peaks that can be related to the presence of GO. In the spectrum of GO, because of extensive oxidation, GO has a strong and broad O-H stretching vibration band at 3410 cm-1, carboxyl C=O stretching band at 1721 cm-1, O-H deformation vibration band at 1404 cm-1 and C-O stretching vibration at 1087 cm-1 (Ciplak Z. et al. Fuller. Nanotub. Carbon Nanostructures 2015, 23, 361-370 and Valencia C. et al. Molecules 2018, 23, 2651).

Line 122 onwards and Fig 4. Description and justification are not evidenced in the findings of Fig 4. This makes relevant statements highy questionable.

The amelioration of using GO, in term of mechanical properties, is visible considering neat chitosan aerogels performance now visible from Figure 4.

Line 139 onwards. Undefined abbreviations.

Thank you, we now defined IC and CBY in the previous sections in which we introduced them.

Line 140 onwards. Undefined experimental procedures, with no clear evidence given in Supplementary Information. No spectrophotometer data is provided.

Thank you for the suggestion, we added information about spectrophotometer in SI section.

Line 155 /166 – Fig. 5-Fig. 6. More descriptive legends are needed on graphs to aid the reader.

We add more information in the figure description to aid the reader.

Line 170 and 185. No porosimetry data is provided.

Porosity was now added, calculated starting from SEM analysis following published procedure (Anguiano M. et al. Plos One 2017, 12, e0171417).

Line 194. Missing data of materials used.

We added information about spectrophotometer.

Line 211 onwards. Data provided is incomplete.

Line 219 section. Data provided is incomplete and not evidenced in Fig. 4

Thank you for the suggestion. We added more details in the manuscript and mechanical properties relative to neat chitosan aerogels.

Line 231 section. The methodology followed is not properly described. No calibration data is mentioned. No spectra is provided,

Thank you for the suggestion, we now introduced the calibration in the manuscript, section 4.6, and additional information about it, with the calibration line plot, in the Supplementary materials.

Line 256. Spectra is mentioned in many sections but, neither correlations stated, nor examples are provided thereof.

We employed Spectra already available in literature for those dyes, we now added the proper reference, even in this section, to help the reader.

General comment: Language and styling need serious revision.  Informal phrases and repetitive sentences like 'thanks...'

Thank you, we revised the manuscript to avoid repetitions and typo or format errors.

Reviewer 2 Report

The manuscript by Pinelli et al. deals with the realization of graphene oxide (GO) /chitosan (CS) composites for dyes removal. The GO/CS composite has been characterized by FTIR, SEM and compression tests. The adsorption capacity of composite towards Indigo carmine (IC) and Cibacron Brilliant Yellow (CBY) has been investigated at different concentrations of pollutants.

In principle, the topic of the work should be interesting for Gels. However, some points need be properly addressed before accepting it for publication:

1) In the introduction it is not exhaustively explained why the authors want to synthesize a GO/CS composite as adsorbent for dyes removal. There is a generic description of the most important properties of GO and CS a review of the well-known state of art (such as mechanical and adsorption properties) of GO/CS composites are not reported. There is a lack of the state of art on the use of GO/CS composites for dyes adsorption!

Therefore, I suggest improving the introduction with more details on why the authors chose to synthesize GO/CS composite for dyes removal and add consistent examples of similar systems already present in literature. See for example:

  • Salzano de Luna, C. Ascione, C. Santillo, L. Verdolotti, M. Lavorgna, G.G. Buonocore, R. Castaldo, G. Filippone, H. Xia, L. Ambrosio. (2019) Optimization of dye adsorption capacity and mechanical strength of chitosan aerogels through crosslinking strategy and graphene oxide addition. Carbohydrate Polymers, 211, 195–203.
  • Banerjee, P., Barman, S. R., Mukhopadhayay, A., & Das, P. (2017). Ultrasound assisted mixed azo dye adsorption by chitosan–graphene oxide nanocomposite. Chemical Engineering Research and Design, 117, 43–56.
  • Qi, L. Zhao, Y. Lin, D. Wu. (2018) Graphene oxide/chitosan sponge as a novel filtering material for the removal of dye from water. Journal of Colloid and Interface Science, 517, 18-27.

2) Why do the authors classify the synthesized material xerogels and not aerogels? A Xerogel is generally obtained when the liquid phase of a gel is removed by evaporation. While, an aerogel is obtained when the liquid from the gel is extracted at supercritical state of the liquid and this is replaced by a gas. Therefore, when a hydrogel is subjected to the freeze-drying process, an aerogels is typically obtained (this is convention nomenclature which is widely adopted).

3) The authors reported the structural, morphological and mechanical characterizations, as well as the absorption capacity of a single system. In my opinion, the work is meager. It is impossible to draw correlations and conclusions.

4) The use of the ammonium persulfate in the formation of the network of the final CS/GO composite is unclear. Can the authors specify (also by adding a reaction mechanism) the chemical interactions of the ammonium persulfate with the GO and the CS?

Author Response

Reviewer 2

The manuscript by Pinelli et al. deals with the realization of graphene oxide (GO) /chitosan (CS) composites for dyes removal. The GO/CS composite has been characterized by FTIR, SEM and compression tests. The adsorption capacity of composite towards Indigo carmine (IC) and Cibacron Brilliant Yellow (CBY) has been investigated at different concentrations of pollutants. In principle, the topic of the work should be interesting for Gels. However, some points need be properly addressed before accepting it for publication:

1) In the introduction it is not exhaustively explained why the authors want to synthesize a GO/CS composite as adsorbent for dyes removal. There is a generic description of the most important properties of GO and CS a review of the well-known state of art (such as mechanical and adsorption properties) of GO/CS composites are not reported. There is a lack of the state of art on the use of GO/CS composites for dyes adsorption!

Therefore, I suggest improving the introduction with more details on why the authors chose to synthesize GO/CS composite for dyes removal and add consistent examples of similar systems already present in literature. See for example:

Salzano de Luna, C. Ascione, C. Santillo, L. Verdolotti, M. Lavorgna, G.G. Buonocore, R. Castaldo, G. Filippone, H. Xia, L. Ambrosio. (2019) Optimization of dye adsorption capacity and mechanical strength of chitosan aerogels through crosslinking strategy and graphene oxide addition. Carbohydrate Polymers, 211, 195–203.

Banerjee, P., Barman, S. R., Mukhopadhayay, A., & Das, P. (2017). Ultrasound assisted mixed azo dye adsorption by chitosan–graphene oxide nanocomposite. Chemical Engineering Research and Design, 117, 43–56.

Qi, L. Zhao, Y. Lin, D. Wu. (2018) Graphene oxide/chitosan sponge as a novel filtering material for the removal of dye from water. Journal of Colloid and Interface Science, 517, 18-27.

Thank you for the suggestion, we used these papers and others to extend the introduction giving the reader an overview of the graphene oxide-chitosan composite systems that can be found in literature.

2) Why do the authors classify the synthesized material xerogels and not aerogels? A Xerogel is generally obtained when the liquid phase of a gel is removed by evaporation. While, an aerogel is obtained when the liquid from the gel is extracted at supercritical state of the liquid and this is replaced by a gas. Therefore, when a hydrogel is subjected to the freeze-drying process, an aerogels is typically obtained (this is convention nomenclature which is widely adopted).

Thank you, we revised the manuscript and made this change in nomenclature.

3) The authors reported the structural, morphological and mechanical characterizations, as well as the absorption capacity of a single system. In my opinion, the work is meager. It is impossible to draw correlations and conclusions.

In this work we investigated the role of adding GO in chitosan aerogels and in particular their use in dye removal from water. The addition of GO is able to increase the mechanical properties obtaining a more stable device, fundamental for their future industrial use. Starting from the synthesis we then characterized the final device, underlining the key role of GO, then applying the system to anionic molecules adsorption. The ability of interact with anionic molecules is typical of chitosan, but the role of graphene oxide is fundamental to improve the performances and the stability of the final device here highlighted.

4) The use of the ammonium persulfate in the formation of the network of the final CS/GO composite is unclear. Can the authors specify (also by adding a reaction mechanism) the chemical interactions of the ammonium persulfate with the GO and the CS?

The formation of the CS/GO gel takes place even in absence of APS, as reported in literature (Chen Y. et al. J. Mater. Chem. A 2013, 1, 1992-2001), on the other hand the characteristics of APS as oxidant agent and initiator are well known. In this work, we employed small amount of APS to promote the crosslinking between the polymeric chains in the system and obtain a more branched and stable framework. We tried to make this clearer with some sentences in the manuscript.

Reviewer 3 Report

Ref. No.: gels-1327737
Title: " Graphene Oxide-chitosan Xerogels: Synthesis, Characterization and Use as Adsorbent Material for Water Contaminants", by F. Pinelli et al., 2021
Journal: Gels

Reviewer Comments to Editor of Gels Journal

This paper focused on preparing the graphene oxide-chitosan xerogels and used as adsorbent material for water contaminants. Although there are some valuable contributions in the experiment and simulate results, it is still not worth to appear in the Gels Journal in its current form. There are some reasons as follows:

  1. Graphene oxide-chitosan xerogels and hydrogels are a popular research subject. But the authors did not survey the related literature enough, some important literature for graphene-based chitosan porous composites should be cited.
  • https://doi.org/10.1021/acssuschemeng.5b00193
  • https://doi.org/10.1007/s40843-015-0090-x
  • doi:10.7569/JRM.2016.634134
  • https://doi.org/10.1002/app.40006
  • https://doi.org/10.1021/sc400352a
  • https://doi.org/10.1016/j.eurpolymj.2019.07.032
  1. The synthesis of graphene oxide-chitosan xerogels has been a mature technology. The characterization measurements presented in the manuscript should add some results for convincing innovation, including TEM, XPS, TGA, BET, or Raman...etc.
  2. The text font in the Figures is too small, especially the text of ordinate and abscissa in Figs 4-6.
  3. Some typo or format error, please check, like “At the end of the day (??)… (Line 64 at page 2), Figure 4. a) (??) Compressive… (Line 136 at page 5), …was purchased from GOgraphene (??) (Line 194 at page 7), we decided to work with the ratio 1:1,7 (??) between (Line208 at page 7).
  4. Add some quantitative results in the Abstract and Conclusion.

In summary, the reviewer thinks this manuscript should be rejected by Gels Journal now. However, if the authors can create more research innovation to meet the quality request of Gels, we can accept the re-submitted manuscript. 

Author Response

Reviewer 3

This paper focused on preparing the graphene oxide-chitosan xerogels and used as adsorbent material for water contaminants. Although there are some valuable contributions in the experiment and simulate results, it is still not worth to appear in the Gels Journal in its current form. There are some reasons as follows:

Graphene oxide-chitosan xerogels and hydrogels are a popular research subject. But the authors did not survey the related literature enough, some important literature for graphene-based chitosan porous composites should be cited.

https://doi.org/10.1021/acssuschemeng.5b00193

https://doi.org/10.1007/s40843-015-0090-x

doi:10.7569/JRM.2016.634134

https://doi.org/10.1002/app.40006

https://doi.org/10.1021/sc400352a

https://doi.org/10.1016/j.eurpolymj.2019.07.032

Thank you for the suggestion, we used some of these papers and others to extend the introduction giving the reader an overview of the graphene oxide-chitosan composite systems that can be found in literature.

The synthesis of graphene oxide-chitosan xerogels has been a mature technology. The characterization measurements presented in the manuscript should add some results for convincing innovation, including TEM, XPS, TGA, BET, or Raman...etc.

Thank you for the suggestion. In this manuscript we are interested in analyzing the advantages of using GO together with chitosan in terms of mechanical properties and final use in dye capture. In this version of the manuscript we added details (like porosity) evaluated starting from SEM analysis and TGA spectrum. XPS and Raman generally do not give more details than IR analysis and this is why they were not considered in this work.

The text font in the Figures is too small, especially the text of ordinate and abscissa in Figs 4-6.

Thank you for the suggestion. The fonts of the axes are now bigger.

Some typo or format error, please check, like “At the end of the day (??)… (Line 64 at page 2), Figure 4. a) (??) Compressive… (Line 136 at page 5), …was purchased from GOgraphene (??) (Line 194 at page 7), we decided to work with the ratio 1:1,7 (??) between (Line208 at page 7).

Thank you, we revised the manuscript to avoid typo and format errors.

Add some quantitative results in the Abstract and Conclusion.

Thank you for the suggestion, we added a couple of sentences both in the abstract and in Conclusion to present some quantitative results.

Round 2

Reviewer 3 Report

The reviewer would like to thank authors for their efforts in amending this revised manuscript according to the reviewer’s most comments. Although not all questions have been answered in this revised version, its quality is still enough to be published in the Gels Journal. I think we can accept this manuscript according to the viewpoint of technological innovation.